



# Field-Test of Wind Turbine by Voltage Source Converter

Nicolás Espinoza and Ola Carlson

Department of Electrical Engineering, Chalmers University of Technology, Gothenburg, SE - 412 96, Sweden

*Correspondence to:* Ola Carlson (ola.carlson@chalmers.se)

**Abstract.** One of the main challenge for the wind energy development is to make the wind turbines efficient in respect of costs while maintaining a safe and reliable operation. An important design criterion is the fulfilment of Grid Codes given by transmission system operators (TSO). The Grid Codes state how wind turbines/farms must behave when connected to the grid in normal and abnormal conditions. In this regard, it is well known that not all the technical requirements can be tested by using

the actual impedance-based testing equipment. For this reason, a new type of testing equipment which comprises the use of fully-rated Voltage Source Converter (VSC) in back-to-back configuration is proposed. Thanks to the full controllability of the applied voltage in terms of magnitude, phase and frequency, the use of VSC-based testing equipment, provides more flexibility as compared with actual testing systems. In addition, the AC grid is decoupled from the tested object when performing the test; meaning that the strength of the grid is not a major limitation. Finally, test results of a 4 MW wind turbine and an 8 MW test

equipment, located in Gothenburg, Sweden, are shown in order to validate the investigated grid code testing methodology.

## 1 Introduction

The reliability of the electrical grid depends on how well the generating units, including wind energy systems, are prepared to support the grid in case of abnormal condition. In this regard, Transmission System Operators (TSOs) have included in their Grid Codes specific technical requirements for interconnection of wind power plants with the electricity grid. In general

words, a Grid Code specifies how a generating plant should behave during normal and abnormal condition of the grid. The continuous increase of electrical energy from wind power injected into the power system has lead TSOs to impose more and more stringent requirements for this kind of plants. For this reason, it is crucial to develop testing methodologies for this type of technology in order to ensure a correct integration of wind energy into the electricity grids (Espinoza, 2016). To evaluate the capability of the wind turbine to withstand grid disturbances, today tests are performed on the generating unit by using an

impedance-based voltage dip generator. By developing further new testing methodologies, it will be possible to test for grid scenarios other than voltage dips, ensuring a reliable and fault-tolerant operation of the wind turbine system. Furthermore, in the future, wind turbines will be required to participate more actively in the regulation of the grid (Tsiliand and Papathanassiou, 2009). In this regard, it is well known that voltage source converters (VSC) can provide the necessary flexibility in order to control the terminal voltage as desired. On the other hand, power electronic devices have become cheaper and more accessible

over the years (Blaabjerg and Ma, 2013). It is, therefore, natural that future testing devices will be fully, or if not, partially driven by VSC devices. For these reasons, a test run has been carried out during January 2015 to August 2017 on Big Glenn



wind turbine (4 MW full-power converter (FPC) wind turbine) by means of a full rated VSC-based testing equipment (8 MW VSC-HVDC).

## 2 Review of Grid Code technical requirements

The Grid Codes considered in this section refer to countries that have high penetration of wind power into their national grid
(EWEA, 2016). Consequently, these countries have developed detailed technical requirements for grid interconnection of wind power plants (Espinoza et al., 2013).

The requirements for steady-state operation of the grid can be mainly categorized in three groups: reactive power requirements for normal voltage operation range; reactive power requirements during nominal active power production; and minimum operation time and active power curtailment during long-term frequency deviations.

These requirements have been compared in Tsiliand and Papathanassiou (2009), Espinoza et al. (2013), Mohseni and Islam (2012) and Altın et al. (2010). A dedicated analysis of the German Grid Code is given in Erlich and Bachmann (2005). Finally, control strategies developed for meeting Grid Code technical requirements have been documented in Bongiorno and Thiringer (2013), Molina et al. (2007) and Martinez and Navarro (2008).

### 2.1 Requirements for steady-state condition of the grid

A TSO can require reactive power injection from the wind farms to support overall system voltage control during normal operation of the grid. Usually, reactive power requirements are delimited inside a minimum power factor range that goes from 0.95 lagging to 0.925 leading and for an active power set-point between 0.05 pu and 1 pu; and within a nominal voltage that varies between 0.9 pu and 1.15 pu.

Reactive power requirement are also dependent of the active power production of the wind farm. For instance, the Danish Grid
Code (ENERGINETDK, 2010) states dependencies between voltage and reactive power, and between active and reactive power production. Both requirements shall be complied simultaneously during normal operation of the wind farm. Moreover, reactive power injection can be controlled by either using a voltage control or power factor control (Tsiliand and Papathanassiou, 2009). An extra option to define the reactive power production is to manually set the reactive power operating point if, for example, a continuous voltage deviation at the connection point is present.

In addition, in Grid Codes it is specified the steady-state frequency and voltage operation range in which the wind turbine should operate continuously. Normal condition is considered for voltages close to 1.0 pu and frequency around 50/60 Hz, with a deviation of approximately ±0.1 pu from the rated voltage and ±0.5 Hz from the rated frequency. Any steady-state grid condition outside these values is defined by a minimum operational time, and in some cases, by a control action on the active power set-point of the wind farm, as enforced by e.g.: the Danish (ENERGINETDK, 2010), Irish (EirGrid, 2015) and
German (E.ON, 2006, 2008) TSOs. When active power curtailment is demanded by the TSO in case of frequency deviations, the generation unit must vary its active power output in order to contribute to the overall regulation of the system frequency. In



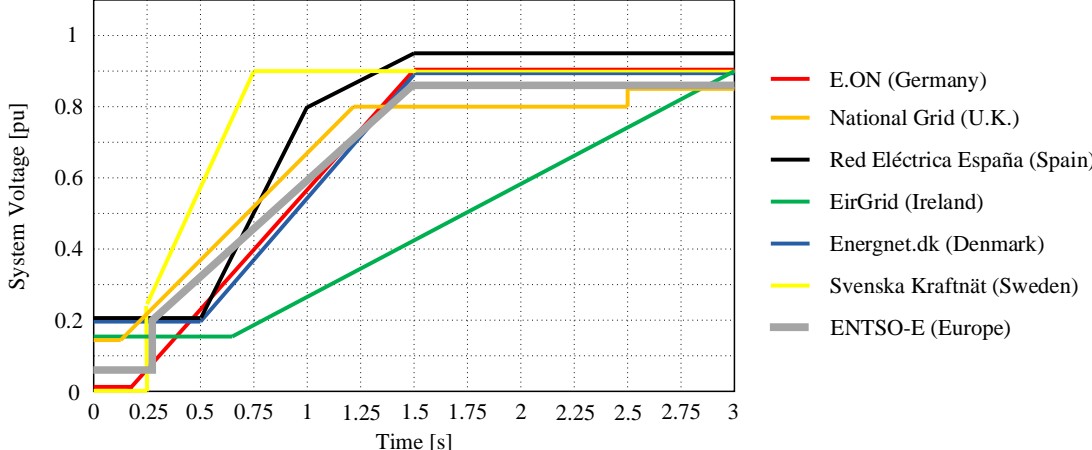

**Figure 1.** Example of LVRT profiles from the selected Grid Codes.

a severe case scenario where the system frequency is outside the frequency operation band, the generating plant is allowed to disconnect.

## 2.2 Requirements for dynamic condition of the grid

In every Grid Code it is specified a voltage dip profile that the wind turbine should ride through without tripping. An exhaustive

5  comparison of LVRT profiles is given in Tsiliand and Papathanassiou (2009). Low voltage ride through profiles characterization in terms of fault time, retained voltage and recovery ramp rates can be found in Mohseni and Islam (2012). In Fig. 1 is shown a combination of the strictest LVRT profiles among the selected Grid Codes. In particular, the European Grid Code ENTSO-E (ENTSO-E, 2016) defines the guidelines to establish the LVRT profiles in each network inside EU.

When it is specified in the Grid Code, wind parks are required to support voltage restoration by injecting reactive power into the

10  grid. In particular, the generating plant must provide voltage support by injecting a specific amount of reactive current during a voltage dip. For example, the Danish Grid Code (ENERGINETDK, 2010) enforces a specific LVRT with retained voltage of 0.2 pu per 500 ms, as shown in Fig. 1, and demands for reactive power support during voltage restoration. Reactive current must be injected when voltage deviates below 0.9 pu. When the system voltage is lower than 0.5 pu, nominal reactive current injection must be reached.

## 15  2.3 Testing for Grid Code fulfilment

The IEC 61400-21 standard issued by the International Electrotechnical Commission (IEC) defines the methodology to test part of the requirement stated in Grid Codes for interconnection of wind turbines (IEC 61400-21 ed2.0, 2008). Moreover, in order to fulfil the standard criteria, specialized testing equipment has been developed (Yang et al., 2012).





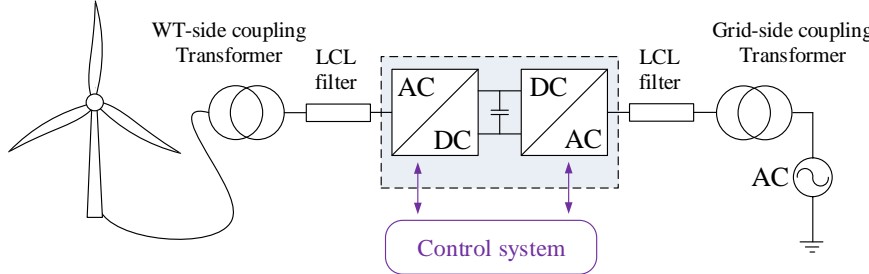

**Figure 2.** Single line diagram of a VSC-based test equipment.

The most common testing device for LVRT test of wind turbines is the impedance-based voltage dip generator (Beeckmann et al., 2010; Martinez and Navarro, 2008; Ausin et al., 2008) . The impedances that constitute the testing device are arranged in order to form a voltage divider at the terminals of the tested object. By a proper selection of these impedances, the amplitude and phase angle of the applied voltage can be controlled as desired. Although it has a simple and robust design which makes

it modular and easily transportable, one of the main drawbacks of this testing device is the fact that it is only able to apply voltage steps variations due to the closing-opening action of the circuit breaker. Moreover, the device is highly dependent of the short-circuit power at the grid connection point, which will also impact the resulting wind turbine voltage (Ausin et al., 2008; Yang et al., 2012; Hu et al., 2009).

Another solution to realize a voltage dip generator is to use fully-rated VSC in back-to-back configuration. By controlling

the turbine-side output of the converter system, the effect of all kind of grid faults can be emulated (Wessels et al., 2010). Thanks to the full controllability of the applied voltage in terms of magnitude, phase and frequency, the use of VSC-based testing equipment, shown in Fig. 2, provides more flexibility as compared with the standard impedance-based test equipment (Espinoza et al., 2013; Ausin et al., 2008), and also brings more advantages in terms of size and weight (Yang et al., 2012; Diaz and Cardenas, 2013). In addition, the AC grid is decoupled from the tested object when performing the test; meaning

that the strength of the grid is not a major limitation, if a proper control strategy of the grid-side VSC of the test equipment is implemented.

Moreover, the LVRT profile given in the majority of the Grid Codes can be fully tested including the recovery ramp (Espinoza et al., 2013), allowing for the emulation of any kind of grid scenario applied to the tested object. Its precise control allows for more possibilities of tests that can be carried out besides what is normally required in the Grid Codes. For example,

frequency characterization of wind turbines can be performed by introducing asynchronous frequency content into the applied voltage while observing the equivalence admittance at the Point of Common Coupling (PCC). Additionally, frequency support capabilities of the tested object can be evaluated by performing a test where the frequency applied at the PCC is varied, as later demonstrated in this paper.

Since wind turbine data is not always available, the testing equipment can be of used to obtain more information from an actual

wind turbine system (Espinoza, 2016). By using a fully controllable VSC-based testing equipment, it is possible to determine





how well the wind turbine is able to reject frequency components other than the fundamental frequency. In this regard, the generating unit can be scanned in a wider frequency range in which the wind turbine can interact with other existing elements present in the interconnected grid. In this regard, frequency scan tests have been carried out by means of the testing device to characterize the input admittance of the tested wind turbine unit. Here, this analysis is limited to the sub-synchronous range

only. The implementation of the controller on the testing equipment is also discussed in Espinoza (2016).

The main drawback of this technology is the fact that is more expensive than using the standard testing device introduced in the previous section. In addition, the control algorithm needed to implement such arrangement of VSC is more complex and extra attention must be given when dealing with over-currents. Moreover, to emulate a grid fault as realistically as possible, a high dynamic performance of the controller that computes the output voltage of the VSC is necessary (Lohde and Fuchs, 2009).

**3   Description of the testing facility**

**3.1   System layout**

The testing equipment is an 8 MW back-to-back HVDC station placed in the harbour of Gothenburg, Sweden. A picture of the actual wind turbine is given in Fig. 3 (Göteborg Energi AB). The wind turbine is located at the edge of the land in proximity to the sea. During the majority of the time, the wind turbine receives offshore wind from the northern part of Denmark. The

station also shown in Fig. 3 houses the interconnecting filters shown in Fig. 3 at the bottom right side. Moreover, a diagram of the layout of the wind turbine connected to the VSC-based testing device is shown in Fig. 2, including interconnecting filters and coupling transformer which are located in the HVDC station.

**Table 1.** Parameters of the field test setup.

| Base values | 10 kV, 4.2 MVA |
|---|---|
| Wind turbine rating | 1 pu |
| Testing equipment rating | 2 pu |
| Type of AC filter in TE | $LCL$ |
| Type of AC filter in WT | $LCL$ |
| Impedance of coupling transformer in testing equipment | 0.08 pu |
| Impedance of step-up transformer in wind turbine | 0.06 pu |
| Interconnection between TE and WT | 300 mts. underground cable |

**3.2   VSC-based testing system description**

In this installation, only the three-phase voltages and three-phase currents at the PCC are sampled by a computer located in the

control room of the HVDC station. The instantaneous active and reactive power are calculated off-line.





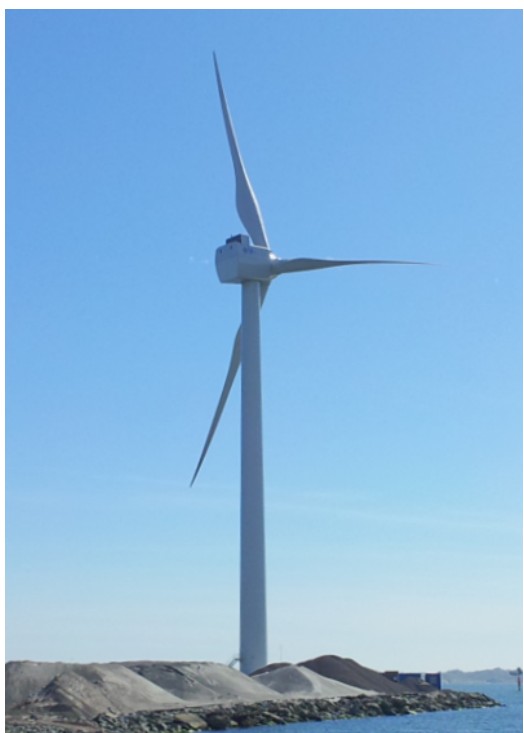

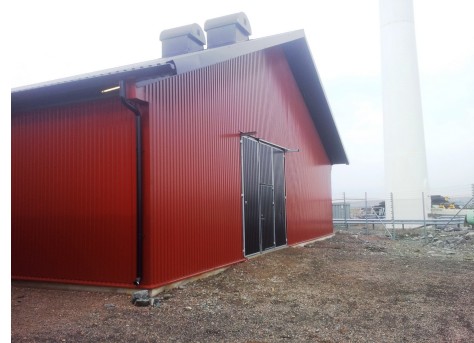

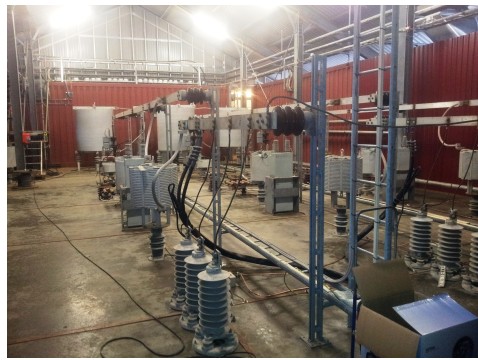

**Figure 3.** Testing facility in Gothenburg comprising the 4.2 MW FPC "Big Glenn" wind turbine; housing of the back-to-back HVDC station, and coupling inductor (back) and AC filters (front) inside the house.

The schematic of the interconnection of the wind turbine with the test equipment is shown in the figure above. The test equipment is rated in 8 MVA, 10.5 kV. The wind turbine is coupled to the testing device through a coupling transformer. The secondary of the transformer is rated at 9.35 kV. A filter bank is placed in order to remove the harmonic content produced by the turbine-side VSC. This converter controls the AC voltage imposed to the wind turbine system, while the grid-side converter is controlling the DC-link voltage by exchanging active power with the interconnecting grid. The testing equipment is interfaced with the grid means of filter bank and coupling transformer, which grid-side is again 10.5 kV. Technical data of the testing setup is given in Table 1.

### 3.2.1 Control of the grid-side VSC

The main blocks that constitute the implemented discrete controller are the DC voltage control and the inner current controller together with the phase-locked loop (PLL). The structure of a typical cascaded controller can be found in Espinoza (2016). The outer DC voltage control generates the reference current for the inner-current controller of the grid-connected converter. The output of the current controller is the reference voltage value for the output voltage of the grid-side VSC. The output reference voltage is sent to a pulse-width modulator (PWM) which computes the gating signals transistors of the VSC (Espinoza, 2016).



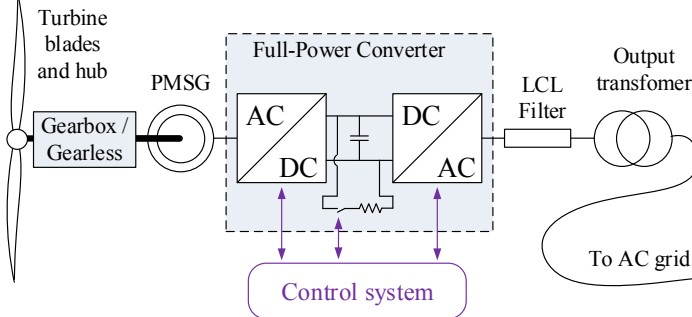

**Figure 4.** Schematic representation of an FPC wind turbine.

### 3.2.2 Control of the turbine-side VSC

In the control algorithm of the turbine-side VSC of the test equipment, a dedicated open-loop voltage control is implemented. The output of the controller is the reference value for the output voltage of the VSC. Finally, a PWM modulator is used to compute the switching signals of the converter (Espinoza, 2016).

### 3.3 Big Glenn wind turbine

In an FPC wind turbine, the generator is connected to the grid through a full-power rated back-to-back VSC, as depicted in Fig. 4. This configuration allows for an increased fault-tolerant capability of the wind turbine, avoiding severe transients in the generator when a grid fault occurs. Moreover, the grid-side converter of an FPC wind turbine can be designed and controlled in order to provide additional reactive power support, without having the need of over-magnetizing the generator core. Moreover, a gearbox is typically used to step-up the rotational speed when coupling the wind turbine hub with the generator shaft. For a direct-drive configuration i.e.: absence of the gearbox in the drive train allowing a direct connection between the hub and the rotor similar to the configuration found in Big Glenn, a dedicated low-speed multi-pole generator must be used in order to achieve the desired nominal frequency in the stator terminals.

The use of fully-rated back-to-back VSC for grid interconnection of wind turbine generators allows for a fast response during abnormal condition of the grid (Beeckmann et al., 2010; Abram Perdana, 2014). For instance, during a voltage dip, the generated power cannot be delivered into the grid due to the absence of sufficient grid voltage. In this scenario, the grid-side-converter can quickly control the current output, avoiding feeding fault currents of large magnitude into the grid. The DC-link capacitor is protected by a DC-crowbar, which allows for the redirection of the produced power into a resistor providing a fast protection during DC over-voltages. The wind turbine is rated in 4.2 MVA, 10.5 kV. The stator of the generator, having a voltage rating of 0.69 kV, is directly connected to the back-to-back VSC. The generator-side VSC, here operated in torque control mode, injects the generated power into the wind turbine DC-link capacitor. The grid-side VSC controls the DC-link voltage by exchanging active power with the imposed AC grid. A filtering stage is placed between the VSC and the LV-side of the wind turbine transformer in order to reduce the harmonic content injected into the electricity grid. Finally, the



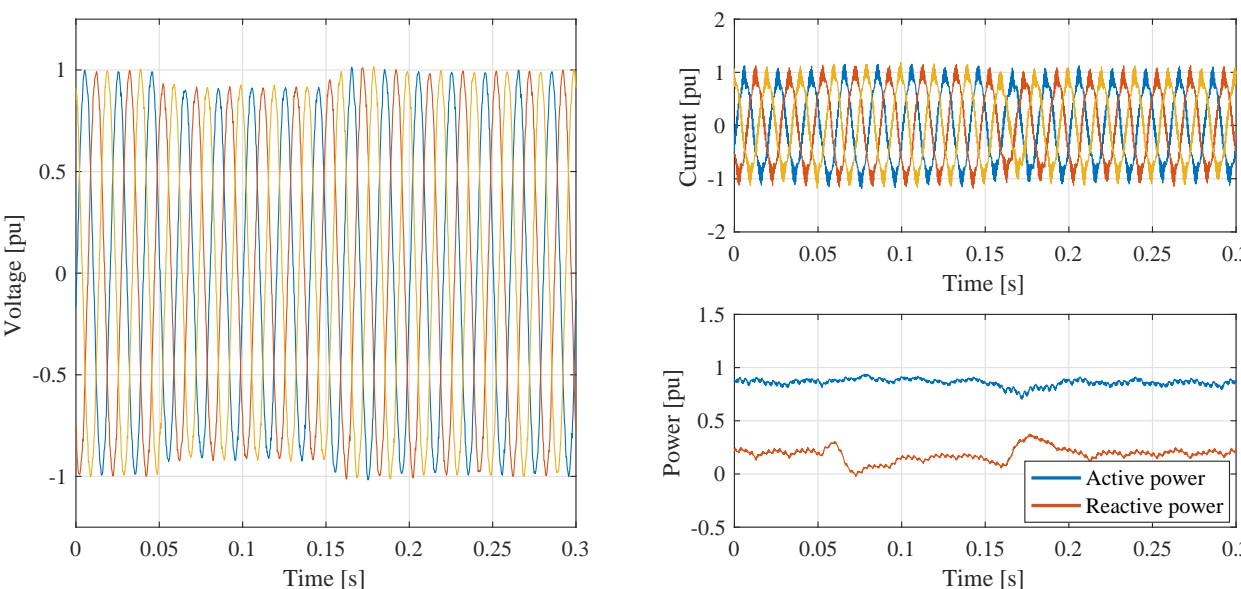

**Figure 5.** Wind turbine tested for voltage dip at full power. In figure: three-phase voltage and current, and active and reactive power output.

output transformer of the wind turbine steps-up the voltage from 0.69 kV to 10.5 kV. Information about wind turbine control during faults can be found in Espinoza (2016), Abram Perdana (2014), Espinoza et al. (2015).

## 4 Tests results

### 4.1 Testing for Low Voltage Ride Through (LVRT)

One of the first tests carried out on the testing facility corresponds to a voltage dip at full power production. The following tests were conducted on January 13th, 2015. The first attempt was to select a relatively small voltage variation, with a smooth transition between normal operation level and retained level. In order to safely conduct the experiment while learning the dynamics of the system, the voltage is reduced from 1 pu to 0.9 pu for 100 ms. The results are shown in Fig. 5.

The three-phase PCC voltage is shown in Fig. 5. At 0.05 s the voltage is reduced from 1 pu to 0.9 pu. At 0.15 s the voltage
is brought back to 1 pu with a ramp function. In order for the wind turbine to maintain constant power production during the voltage reduction, the generating unit increases the magnitude of the current, while maintaining a constant power production, as also depicted in Fig. 5. During the voltage variation, the wind turbine maintains its active power set-point, injecting a steady 0.9 pu of active power into the imposed grid. From the figure, it is possible to observe that the wind turbine does not engage its LVRT control. Thus, the reactive power is reduced only due to the reduction of the voltage across the AC-link between the
wind turbine and the testing equipment.

A second test has been carried out the same day. Here, the voltage is varied from 1 pu to 0.65 pu for 100 ms while the wind turbine is producing nominal active power. The results are depicted in Fig. 6. The three-phase PCC voltage is dropped at t=





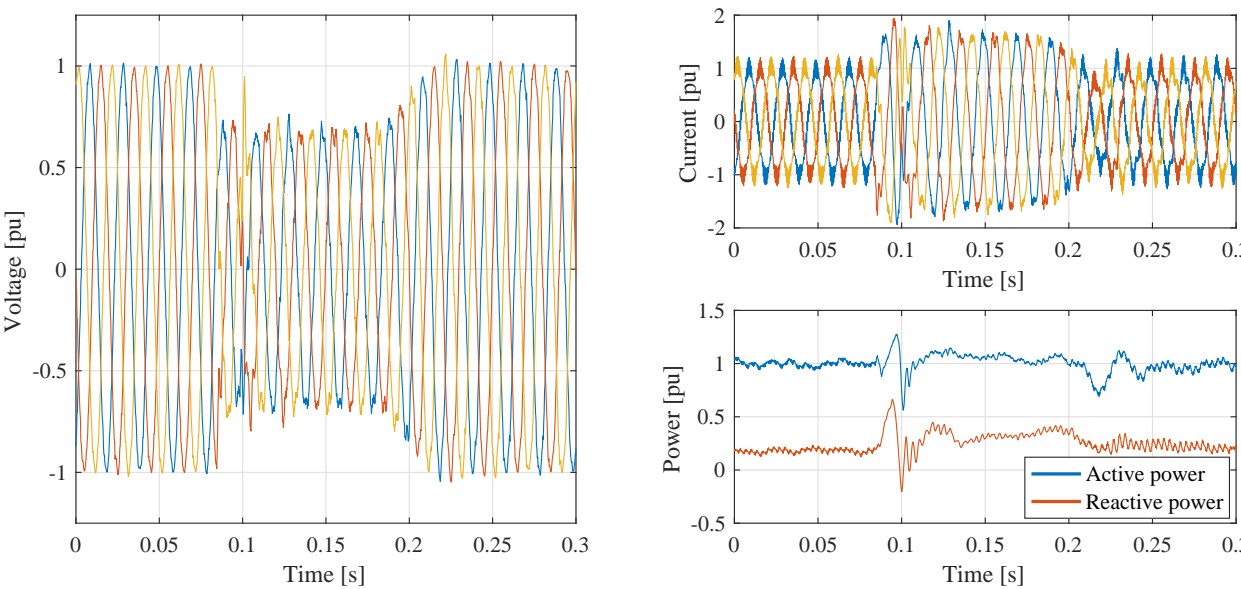

**Figure 6.** Wind turbine tested for LVRT at full power. In figure: three-phase voltage and current, and active and reactive power output.

0.05 s. In this scenario, a stiff ramp rate of 0.2 pu/ms has been selected for the voltage dip while a ramp of 0.0125 pu/s has been set for the recovery. At 0.075 s it is possible to observe a transient in both the current and the voltage shown in the figure. The transient lasts for 7 ms and the wind turbine manages to control its current output during the voltage dip.

The wind turbine active power set-point is maintained at 1 pu, while the reactive power is boosted reaching a mean value of

0.4 pu during the voltage dip. Finally, it is possible to observe that the pre-fault reactive power exchange at the measurement point is 0.2 pu. The wind turbine injects an additional 0.2 pu of reactive power when detecting the voltage dip at its terminals. Therefore, a total of 0.4 pu is maintained until the voltage is increased back to 1 pu.

The voltage starts to recover at 0.15 s with a ramp function. Observe that the current is also reduced at the same time that the reactive power is brought back to 0.2 pu. The active power oscillates at 0.18 s while the voltage has reached a steady-state

level of 1 pu. Observe that the current is above 1 pu during the voltage dip, meaning that the wind turbine have over-current capabilities and it is capable to momentarily increase the output current beyond nominal values during the voltage dip.

### 4.2   Reactive power control during voltage dip

A similar test has been carried out on the 17th of May of 2016. Unfortunately, the wind turbine was operating at low wind speed. However, the lack of produced active power made the variations in the reactive power more prominent, as later see in

this section. This experiment consists in a voltage variation from 1 pu to 0.75 for about 200 ms. In order to avoid an oscillatory response similar to the ones experienced in the previous experiment, the down-slope ramp of the controlled VSC voltage is set at 0.02 pu/ms and the recovery ramp is set at 0.01 pu/ms. From the voltage waveform given in Figure 10, it is possible to observe that the voltage is controlled by the testing equipment in an effective and smooth way.





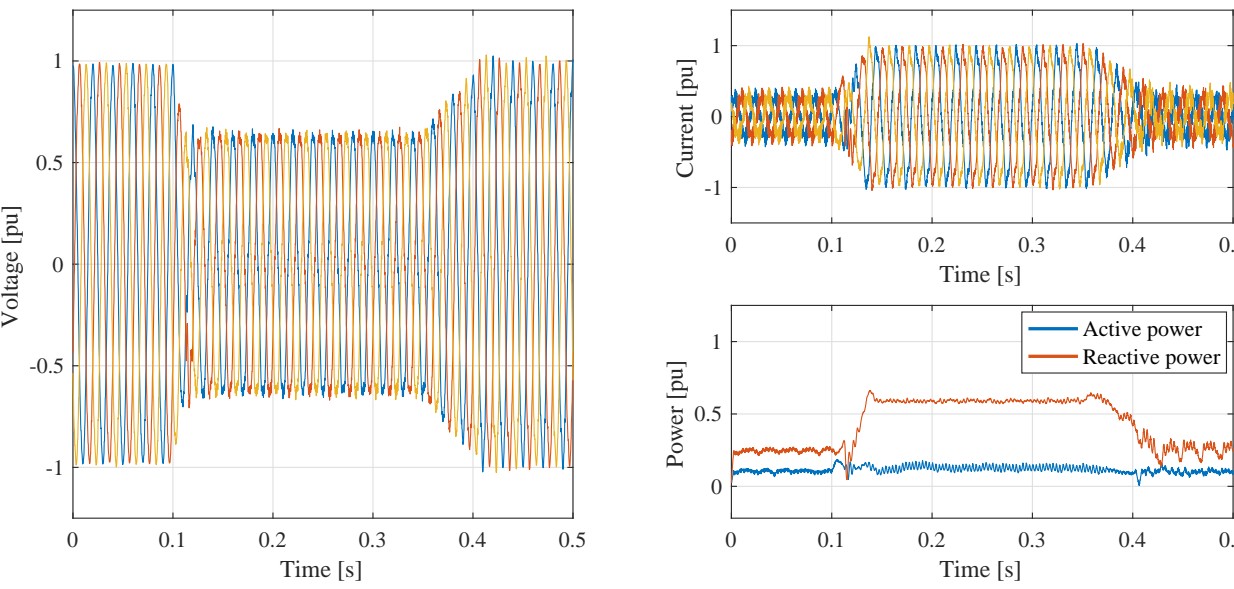

**Figure 7.** Wind turbine tested for LVRT at low power production. In figure: three-phase voltage and current, and active and reactive power output.

The active power was set to 0.15 pu during this test. Moreover, the reactive power seen in Fig. 7 is mainly due to the capacitor bank of the local LCL filter placed at the terminals of the turbine-side VSC of the testing equipment, while the variation observed during the voltage dip is due to the control action of the wind turbine.

When the voltage dip is detected at 0.12 s, the wind turbine injects reactive current. As shown in the figure above, the current reaches 0.95 pu, meaning that the turbine is operated in proximity to the rated current. In order to quickly boost the voltage, the reactive power shown also in Fig. 7 is increased with a ramp function. There is a small overshoot in the current that is reflected on the reactive power at 0.14 s, mainly due to the fact that the voltage has reached steady-state during the dip while the current continues to increase. This can be attributed to the voltage monitoring system and the reactive power controller of the wind turbine during dynamic condition of the grid.

The reactive power injection is maintained for the complete duration of the voltage dip. The (mainly reactive) three-phase current is later reduced when the voltage is restored to 1 pu. At 0.31 s, a small overshoot in the reactive power is experienced. Note that the current is maintained during a short period of time at 0.85 pu, while the voltage has already started to increase towards 1 pu. The system reaches a post-fault steady-state at 0.43 s. It can be observed that there is a small transient on the reactive power set-point after the voltage dip. This can be attributed to the wind turbine control action when calculating a reactive current reference based to a varying instantaneous measured voltage. Afterwards, the wind turbine will resume normal operation, reducing the current (and therefore the reactive power output) to its pre-fault operating point.





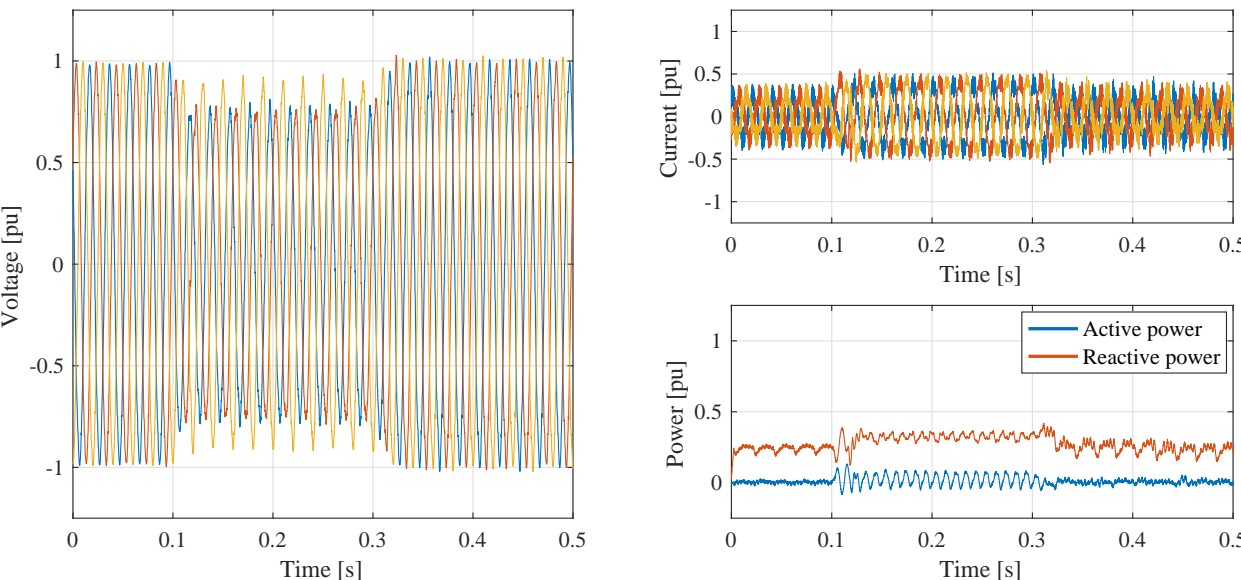

**Figure 8.** Wind turbine tested for unbalanced voltage dip at low power production. In figure: three-phase voltage and current and active and reactive power output.

### 4.3 Testing for unbalance voltage dip

In this section the response of the wind turbine under unbalanced voltage dip is studied. This test was carried out again on the 17th of May of 2016, at low wind conditions. The turbine-side VSC of the actual testing equipment is controlled in open-loop and the voltage in phase a and b are dropped from 1 pu to 0.7 pu for 200 ms. The voltage in phase c is maintained at 1 pu during all the duration of the test. The resulting PCC voltage is given in Fig. 8 . Observe that the reactive power (green traces) is increased from 0.25 pu to 0.35 pu approximately, while a relatively small oscillation at 100 Hz in both active and reactive power is seen. The distorted current shown in Fig. 8 accounts for oscillations seen in the power, although these oscillations are small.

These last three examples show in a clear way the response of the wind turbine, particularly the reactive power controller during a balanced and unbalanced voltage dip. Considering that the control system of the tested wind turbine is unknown, these results suggest that the controller implemented on the wind turbine accounts also for the negative sequence, while still experiencing small oscillations in its output power.

### 4.4 Impact of the voltage dip in wind turbine converter

A voltage dip applied at the terminals of the VSC of the testing equipment affect drastically the voltage of the wind turbine converter. Moreover, the controller of the wind turbine VSC will also react against the voltage dip measured at the LV–side of its output transformer (where the local LCL filter is shown in Fig. 4). Although the voltage will be somewhat smooth




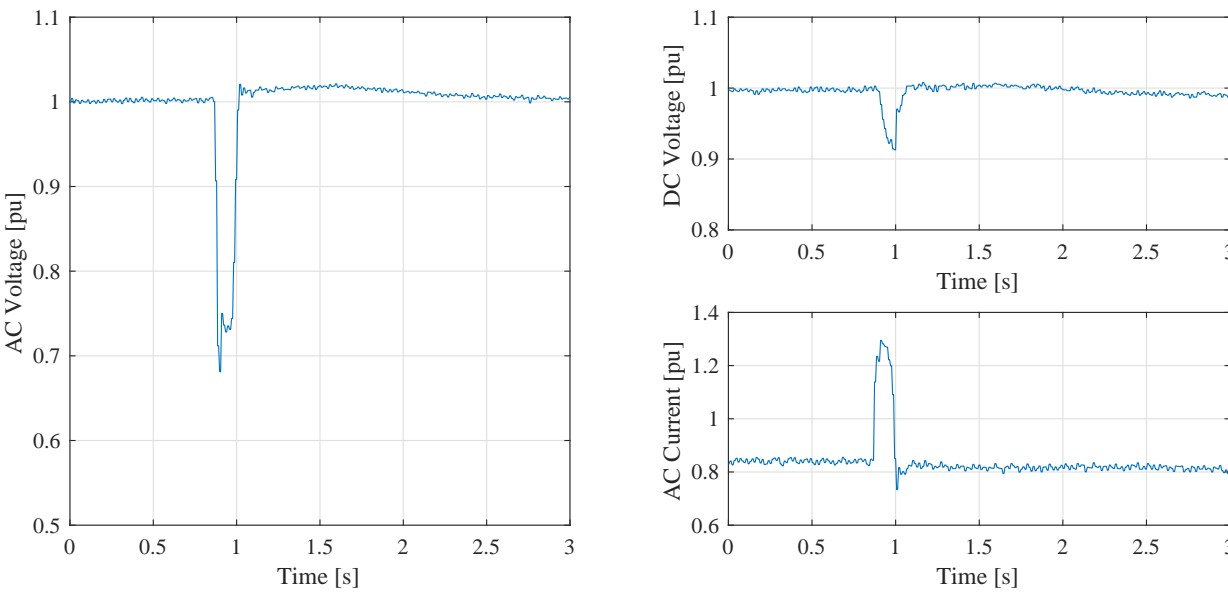

**Figure 9.** Wind turbine VSC under a non-severe voltage dip. In figure: AC voltage and current magnitude in LV-side of wind turbine converter, dc voltage and power output of one of the wind turbine converters.

because of the filtering action of the overall inductance, the resulting output current and DC voltage of the back-to-back wind turbine transformer are controlled in such way that the wind turbine can maintain normal operation as much as possible. In the following, the internal signal of the wind turbine converter are plotted for some of the cases given the previous section, and the effect of the voltage dip in further discussed. For a better understanding of the dynamics inside the wind turbine converter,

the physical magnitudes of speed, voltage and power have being kept, while the measured torque and mechanical power at the generator are shown in percentage of their nominal values.

If the voltage reduction is not dip enough, the wind turbine might have some extra room in its rating in order to maintain normal power production at a reduced voltage (see Fig. 5). This can be done by increasing the current immediately after detecting a voltage dip. In Fig. 9 it is possible to observe the effect of a voltage dip in the wind turbine converter. The voltage imposed is

10 measured at 1 pu and is dropped to 0.74 pu for 150 ms approximately. The line current is also plotted in Fig. 9 and it increases fast when the voltage dip is detected at 0.08 s. The pre-fault value is 0.85 pu and rises to 1.3 pu during the dip.

The DC voltage also shown in Fig. 9 is dropped during the test. This can be attributed to the fast increasing in the wind turbine current, which can be faster than the time constant of the DC link capacitor, allowing a normal power flow during the dip while affecting slightly or even decreasing the DC voltage.

Finally, it is interesting to see the decoupling that exist between the grid-side VSC and the generator-side VSC of the wind turbine. In Fig. 10 it is shown the generator torque and generator speed when the grid voltage experiences a dip. If the conditions are met so that the wind turbine VSC response is smooth, the generator is not affected. Here the torque is maintained constant





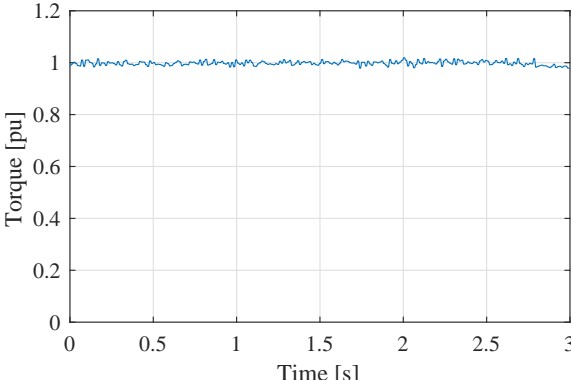
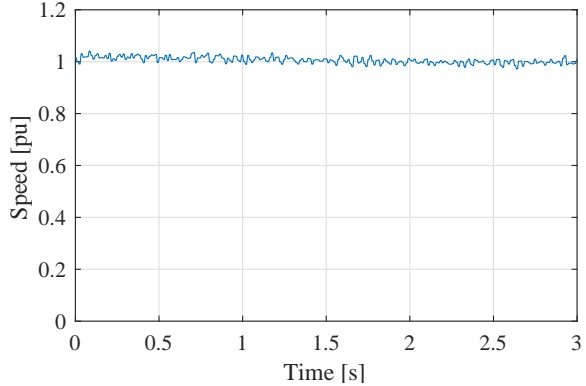

**Figure 10.** Wind turbine VSC under a non-severe voltage dip. In figure: generator torque and generator speed.

at 1 pu and the speed is controlled at 1 pu as well, meaning that the wind turbine system is operated at full power. Note that the wind turbine converter have over-current capabilities in order to maintain a constant power flow during transients.

### 4.5 Testing for frequency deviation

In this case study, the voltage applied to the 4 MW wind turbine is controlled at 1 pu and only the frequency is varied. The obtained results are given in Fig. 11 . In order to measure for a long period of time, the voltage and the current are here

measured by a portable measurement unit located upstream of the LCL filter. In particular, at the converter-side of the coupling transformer placed at the turbine-side VSC-HVDC station shown in Fig. 3. For this reason, the reactive power curve depicted in red in Fig. 11 has a different value with respect to the other case studies shown throughout this chapter.

Here, only the measured frequency and the active and reactive power output are shown. The first scenario corresponds to a frequency drop of 0.5 Hz for 15 seconds. From the first case shown in Fig. 11 it can be noticed that the frequency is varied with

a ramp of 0.2 Hz/s, or 5 seconds per varied Hz, for both the drop and the recovery of the frequency. According to the Swedish Grid Code (Svenska Kraftnät, 2005), the wind turbine should maintain its active power production at any given frequency within the normal frequency range (49.5 Hz to 50.5 Hz); while varying its output power for frequencies outside this range and ultimately cease to operate at frequencies outside the full operational range (48 Hz to 52 Hz) . The upper plot in Fig. 11 for frequency dip shows that the active power is kept constant at 0.7 pu for the majority of the test. At 30 s the active power is

slightly reduced, mainly due to the variations in the wind speed at the moment of the test.

The second scenario corresponds to two consecutive frequency swells of 1 Hz. The frequency is initially controlled at 50 Hz and varied upwards with a ramp of 0.05 Hz/s, or 20 seconds per varied Hz. A frequency of 51 Hz is maintained for 25 seconds approximately. Afterwards, the frequency is increased to 52 Hz. The results are shown in the right-side of Fig. 11 .

Observe in the lower plot that the upwards and downwards tendency of the output power suggest that the wind turbine is varying

its operating point according to the wind speed and not in demand of the applied frequency. In addition, the active power is slightly increased 10 seconds after the frequency reaches 51 Hz, at 40 s, while continues to increase with the increasing of the





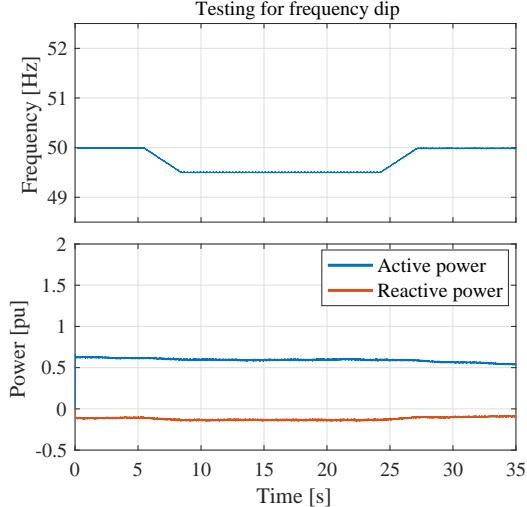
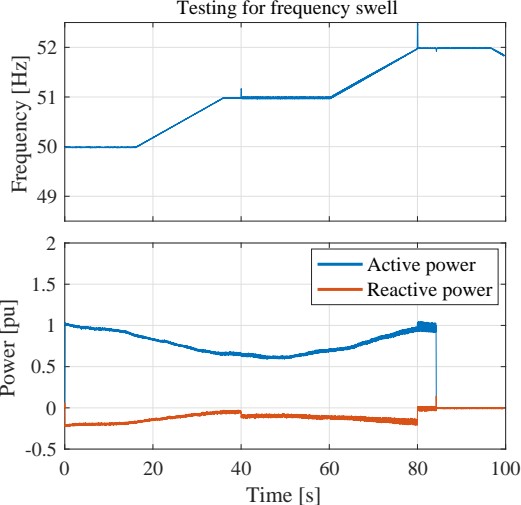

**Figure 11.** Frequency variation test. In figure: set-points of active and reactive power during frequency dip (left-side figures) and frequency swell (right-side figures).

system frequency at 60 s. Finally, a critical point is encountered at 80 s when the frequency reaches 52 Hz. The wind turbine enters into an operation mode that affects the active power output while experiencing an oscillation at 104 Hz. The wind turbine shuts down by an over-frequency protection relay, 5 seconds after the frequency reaches 52 Hz, at t=85 s. The different power production levels experienced when performing the test are also somewhat reflected on the reactive power output of the wind turbine, as seen in green traces for both scenarios shown in Fig. 11 .

**4.6 Frequency scan**

This section shows first the frequency scan test carried out on the 20th of June, 2016, when the wind turbine was operating at 0.6 pu power of production. Afterwards, the result from the frequency scan test, carried out on the 17th of May, 2016, at low power production, is given. The current and voltage are measured at the HVDC station, where the voltage applied to the wind turbine is controlled. The frequency scan is performed by adding a voltage component of magnitude 0.015 pu modulated
at the interested frequency, on top of the fundamental reference voltage of 1 pu. The measured phase voltage and line current are evaluated at the frequency of interest by using the methodology given in Espinoza (2016), also reported in Espinoza et al. (2016).

The wind turbine is changing its operating conditions during each tested frequency and these are shown in Fig. 12 in bars. The average output power of the wind turbine is of 2.9 MW. However, a wide variation between 2.6 MW and 3.5 MW has
15 been encountered during the test. The wind turbine impedance $Z_\mathrm{w}(j\omega)$ is obtained as an average of the phase impedances. The admittance $Y_\mathrm{w}(j\omega)$ can be calculated by the inverse of $Z_\mathrm{w}(j\omega)$. The real and imaginary part together with the magnitude of the measured phase admittance $Y_\mathrm{w}(j\omega)$ are shown in Fig. 12 with blue and green dots respectively. The magnitude of the



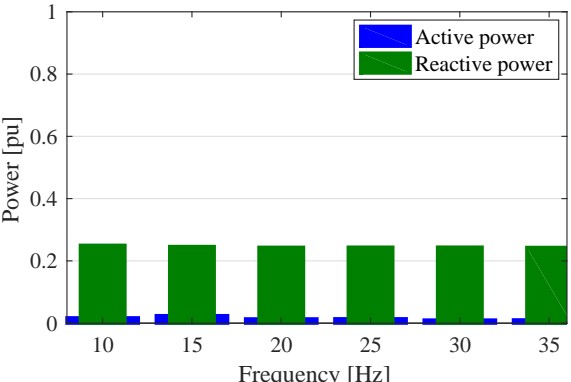 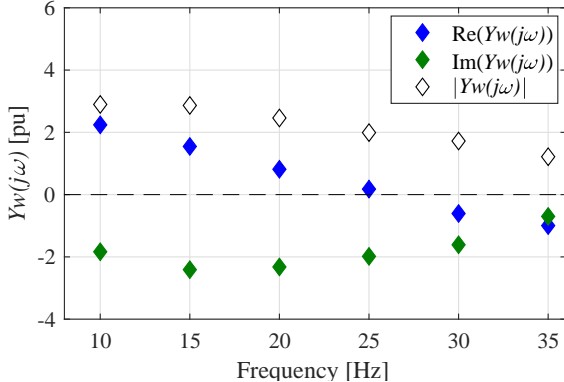

**Figure 12.** Results of frequency scan on the wind turbine operated at 5% of power production. In figure: operating points prior the scan; measured admittance components.

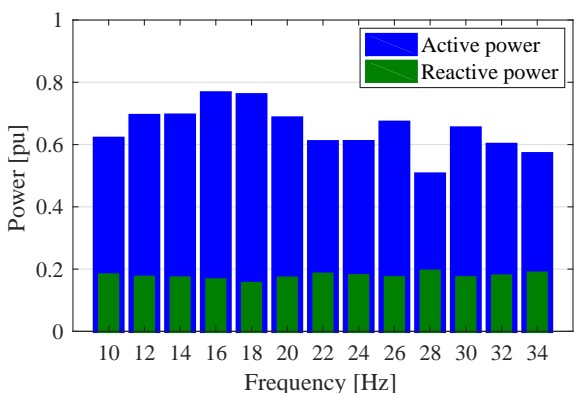 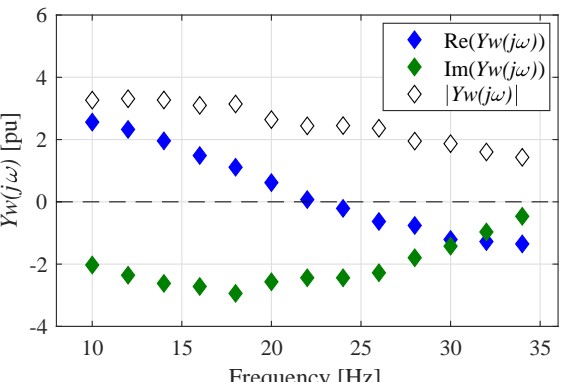

**Figure 13.** Results of frequency scan on the wind turbine operated at 65% of power production. In figure: operating points prior the scan; measured admittance components.

admittance is plotted in white dots. The resulting measured points in Fig. 12 suggest that the wind turbine presents a positive real part for frequencies below 22 Hz, with a maximum measured value of 2.8 pu at 10 Hz. On the other hand, a relatively high non-passivity behavior is exhibit for frequencies above 30 Hz, meaning that the wind turbine could resonate if these frequencies are encountered in the network. The minimum value for $Re(Y_{\mathrm{w}}(j\omega))$ is -1.5 pu and can be observed at the last scanned frequency of 34 Hz.

5    The scanned imaginary part of the admittance $Im(Y_{\mathrm{w}}(j\omega))$ is also shown in Fig. 12 in green dots. The turbine seems to present a capacitive behavior for most of the studied frequency range. The reactive power set-point at the measurement point is dependent on both the configuration of the filtering stage at the terminals of the VSC-HVDC and on the wind turbine reactive power controller. The minimum value of $Im(Y_{\mathrm{w}}(j\omega))$ is found to be -2.4 pu at 18 Hz, exhibiting its maximum capacitance in the scanned frequency range. For frequencies above 20 Hz, $Im(Y_{\mathrm{w}}(j\omega))$ increases up to a maximum of -0.4 pu measured at





10  34 Hz, corresponding to its minimum capacitance in the synchronous range. Observe, however, that reactive power set-point shown Fig. 12 is kept relatively constant at 0.2 pu (0.9 MVAr) at 50 Hz during all the test.

The absolute value of the measured wind turbine admittance $|Y_{\mathrm{w}}(j\omega)|$ is also shown in in Fig. 12 in white dots. The operating point of the wind turbine at the moment of each test impacts both the amplitude and also the vector components of its input admittance. For this reason, the uneven trend of $|Y_{\mathrm{w}}(j\omega)|$ somewhat matches the variation on the output power of the wind

turbine at each scanned frequency.

Finally, one last frequency sweep is given in Fig. 13. The test was carried out in a very low windy day; therefore, the wind turbine is operated at low power production. The figure shows the active and reactive power set-points at the moment of each test. Moreover, it is possible to note that the power flow is mainly dominated by reactive power coming from the filter banks between the wind turbine and the testing equipment. The measured reactive power is 0.023 pu for all the test. The calculated

admittance components are shown in dots also in Fig. 13.

The real part of the admittance $Re(Y_{\mathrm{w}}(j\omega))$, show in blue dots in Fig. 13, is slightly reduced to 2.2 pu, as compared to 2.8 pu given for the previous case. The main difference is the zero-crossing of the real part, which in this case occurs at 26 Hz, instead of 23 Hz for mid-power operation. From the plots where the reactive power is given, this second tests shows a small increase in the reactive power measured at the HVDC station. Thus, the imaginary part of the measured admittance can be affected by

the operating point of the wind turbine, making the green trace in Fig. 13 slightly closer to 0 pu, as compared with Fig. 12. Finally, the total magnitude $|Y_{\mathrm{w}}(j\omega)|$ of the admittance is decreased from 4 pu from the first test, to 2.7 pu when operated at no power, as shown in white dots in the figure. Observe that the trend of $|Y_{\mathrm{w}}(j\omega)|$ is somewhat smoother given that the wind turbines operates a low power during the complete duration of the scan.

## 5   Conclusions

In this paper it has been demonstrated that the full characterization of the wind turbine system can be carried out by the use of a flexible VSC-based test equipment. The full controllability of the testing device allows for testing of multiple grid scenarios, making it possible not only to determine the behavior of the generating unit against common grid contingencies, but also to evaluate the performance of the generating unit in further improving the overall reliability of the grid. This includes, for example, the evaluation the operating modes that are of interest for the stability of the interconnected power system.

Field test results of Big Glenn 4 MW wind turbine and 8 MW testing equipment have been included in this paper. The tests carried out on the actual wind turbine system include balance and unbalanced voltage dips, defined by different retained voltage and different ramp-rates, as well as frequency variation tests and frequency scan. The results shows that a LVRT control strategy is implemented on the tested system injecting reactive power when a voltage dip is detected. Moreover, it has been shown that the generating unit maintains a smooth control of the reactive power output during unbalanced voltage dips, at least for low

power set-points. These results demonstrate that a VSC-based testing device can be used to evaluate how well a wind turbine system can withstand the technical requirements given in the Grid Codes.



The multi megawatt FPC wind turbine system has also been characterized in the sub-synchronous range by means of the interconnected VSC-based testing equipment. The frequency scanning technique has been demonstrated by field test and the input admittance of the generating unit has been evaluated for two operating conditions. The frequency trend of the scanned turbine exhibits a non-passive behavior at higher frequencies within the sub-synchronous range while also exhibiting capacitive behavior throughout the whole scanned range. The unique field tests presented in this report have provided an experimental validation of the proposed wind turbine testing methodology, particularly on the wind turbine impedance characterization and on the evaluation of its steady-state and dynamic performance against different conditions of the grid.

*Acknowledgements.* This work has been carried out within the Swedish Wind Power Technology Centre (SWPTC). The financial support provided by the academic and industrial partners including Göteborg Energi, Protrol, the Västra Götaland Region and the Swedish Energy Agency is gratefully acknowledged.



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
