# Peer review of "Field-Test of Wind Turbine by Voltage Source Converter"

_Wind Energy Science, 2018_

## Referee Comment (RC1) · H. Polinder (Referee) · 14 Jul 2018

This paper presents test results of new method to test if a wind turbine meets the grid codes using a voltage source converter. What I appreciate very much is that this paper does not report on more simulations (as many other publications do), but on a test setup that has been built to test wind turbines. Building a voltage source converter with a power level of 8 MW and controlling it with such dynamics that it can simulate grid faults is a huge engineering job. This setup makes it possible to do lots of other tests that the current test setups using voltage dividers cannot do.

The authors presented this idea in earlier publications. This paper presents test results that show the equipment works.

For me, the most important question is: what is the scientific contribution of this paper? The authors indicate the development of testing technology is necessary for further integration of renewables. Maybe, it helps if the authors explain what makes this converter different from a standard converter. I would appreciate a paper that describes the concept, the design, the implementation and the testing together more; this paper just shows extensively the VSC works, and that could be done shorter.

I have a number of questions/comments:

1 More than half of the paper presents test results: measurements of voltages, currents and powers under different circumstances. Can the test of fig 5 be omitted because it does not add anything to fig 6?

2 Doing a frequency scan is something that is possible with a VSC. However, why is that useful? What can we do with the results?

3 p9, line 1, I think the ramp is 12.5 pu/s instead of 0.0125 pu/s. Is that correct?

4 I miss a reference to what seems to be a similar paper: C Saniter; J Janning, "Test Bench for Grid Code Simulations for Multi-MW Wind Turbines, Design and Control", IEEE Transactions on Power Electronics, 2008, Volume: 23, Pages: 1707 - 1715. What does the paper under review add to this one?

5 There are too many language mistakes. A number of examples:

p3 line 4: In every grid code it is specified => Every grid code specifies

p3 line 6: In fig. 1 is shown => Fig. 1 shows

p4 line 7: dependent of => dependent on

p4 line 24: can be of used to obtain => can be used to obtain

p7 line 10-13: unclear sentence

p9 line 14: as later see =>??
p12 line 3: signal of the wind turbine converter are => signals of the wind turbine converter are

p12 line 5: have being kept => have been kept

p12 line 7: dip => deep

---

## Referee Comment (RC2) · Anonymous Referee #2 · 25 Feb 2019

Oberall a nice and relevant paper. I agree with the comments of the first author, further more I recommend to explain all used shortcuts like pu LVRT ...

---

## Author Comment (AC1) · 22 Mar 2019

This paper presents test results of new method to test if a wind turbine meets the grid codes using a voltage source converter. What I appreciate very much is that this paper does not report on more simulations (as many other publications do), but on a test setup that has been built to test wind turbines. Building a voltage source converter with a power level of 8 MW and controlling it with such dynamics that it can simulate grid faults is a huge engineering job. This setup makes it possible to do lots of other tests that the current test setups using voltage dividers cannot do. The authors presented this idea in earlier publications. This paper presents test results that show the equipment works. (1) comments from Referees: 1. More than half of the paper presents test results: measurements of voltages, currents and powers under different

circumstances. Can the test of fig 5 be omitted because it does not add anything to fig 6?

(2) author's response As explained in the text the LVRT control is not activated in the small dip test in fig 5 whereas in fig 6 the LVRT mode is activated and the control of active and reactive power starts a small oscillation in active and reactive power. And this can be a message for even larger transients if the dip is lager. For this reason, we believe this case must stay in the paper.

(3) author's changes in manuscript. A note on this will be added to the paper.

---

## Author Comment (AC2) · 22 Mar 2019

(1) comments from Referees: Doing a frequency scan is something that is possible with a VSC. However, why is that useful? What can we do with the results?

(2) author's response: This information is vital when, for example, analysing wind farm system data to identify the risk for sub-synchronous oscillations. This information is also valuable to evaluate the performance of large time-constant controller such as voltage control or power oscillations damping control that can be implemented in a wind turbine system or at wind farm level.

(3) author's changes in manuscript. A note on this will be added in the paper for better understudying of the reader.

---

## Author Comment (AC3) · 22 Mar 2019

(1) comments from Referees 3: p9, line 1, I think the ramp is 12.5 pu/s instead of 0.0125 pu/s. Is that correct?

(2) author's response, Thank you for pointing this out. There is a m missing in pu/s, pu/ms.

(3) author's changes in manuscript. It will be corrected in the next revision.

---

## Author Comment (AC4) · 22 Mar 2019

(1) comments from Referees: I miss a reference to what seems to be a similar paper: C Saniter; J Janning, "Test Bench for Grid Code Simulations for Multi-MW Wind Turbines, Design and Control", IEEE Transactions on Power Electronics, 2008, Volume: 23, Pages: 1707 - 1715. What does the paper under review add to this one?

(2) author's response: Thank you for the suggestion. The abstract indicate that that paper describe the test setup and not the use of it in a real test with a large wind turbine

(3) author's changes in manuscript: A discussion will be added to our paper and the list of reference will be updated.-.

---

## Author Comment (AC5) · 22 Mar 2019

(1) comments from Referees: There are too many language mistakes. A number of examples:

(2) author's response, Thank you for pointing this.

(3) author's changes in manuscript. The paper will be checked for proofread

————————————————

---

## Author Response (AR1)

Dear reviewer and editor,

We want to start by thanking you for the positive general comments and good specific suggestions that we can use to improve the paper. We will address the comments below.

Questions from reviewer Henk Poulinder

This paper presents test results of new method to test if a wind turbine meets the grid codes using a voltage source converter. What I appreciate very much is that this paper does not report on more simulations (as many other publications do), but on a test setup that has been built to test wind turbines. Building a voltage source converter with a power level of 8 MW and controlling it with such dynamics that it can simulate grid faults is a huge engineering job. This setup makes it possible to do lots of other tests that the current test setups using voltage dividers cannot do.
The authors presented this idea in earlier publications. This paper presents test results that show the equipment works.

1. More than half of the paper presents test results: measurements of voltages, currents and powers under different circumstances. Can the test of fig 5 be omitted because it does not add anything to fig 6?

2. Doing a frequency scan is something that is possible with a VSC. However, why is that useful? What can we do with the results?
3 p9, line 1, I think the ramp is 12.5 pu/s instead of 0.0125 pu/s. Is that correct?

4. I miss a reference to what seems to be a similar paper: C Saniter; J Janning, "Test Bench for Grid Code Simulations for Multi-MW Wind Turbines, Design and Control", IEEE Transactions on Power Electronics, 2008, Volume: 23, Pages: 1707 - 1715. What does the paper under review add to this one?

5. There are too many language mistakes. A number of examples:
p3 line 4: In every grid code it is specified => Every grid code specifies
p3 line 6: In fig. 1 is shown => Fig. 1 shows
p4 line 7: dependent of => dependent on
p4 line 24: can be of used to obtain => can be used to obtain
p7 line 10-13: unclear sentence
p9 line 14: as later see =>??
p12 line 3: signal of the wind turbine converter are => signals of the wind turbine converter are
p12 line 5: have being kept => have been kept
p12 line 7: dip => deep

Replay:

Dear Henk,

Thanks for the review, some comments follows, we have also updated the paper for most of your comments. Especially we have in a more clear way explained the benefits with frequency scan and improved the paper linguistically.

1. As explained in the text the LVRT control is not activated in the small dip test in fig 5 whereas in fig 6 the LVRT mode is activated and the control of active and reactive power starts a small oscillation in active and reactive power. And this can be a message for even larger transients

if the dip is lager. For this reason, we believe this case must stay in the paper. A note on this will be added to the paper.

2. This information is vital when, for example, analysing wind farm system data to identify the risk for sub-synchronous oscillations. This information is also valuable to evaluate the performance of large time-constant controller such as voltage control or power oscillations damping control that can be implemented in a wind turbine system or at wind farm level. A note on this will be added in the paper for better understudying of the reader.

3. Thank you for pointing this out. There is a m missing in pu/s,  pu/ms. It will be corrected in the next revision.

4. Thank you for the suggestion and for the interest in the topic. The abstract indicate that that paper describe the test setup and not the use of it in a real test with a large wind turbine A discussion will be added to our paper and the list of reference will be updated. In the new text can be read: "For example, the test setup presented in Saniter and Janning (2008) consists of a complex configuration of several VSCs and a three-winding transformer. The paper discusses a comparison between no-load tests and simulation results, while the effectiveness of actual tests remain"

5. Thank you for pointing this. The paper will be checked for proofread.

The editor also like us to explain the scientific value of the paper:

This paper has demonstrated that the full characterisation of the wind turbine system can be carried out by using flexible VSC-based test equipment. The full controllability of the test device allows for testing of multiple grid scenarios, making it possible not only to determine the behaviour of the generating unit against common grid contingencies, but also to evaluate the performance of the generating unit in further improving overall grid reliability. This includes evaluating different operating modes of the wind turbine which can be of interest for the overall stability of the interconnected power system.

The unique field tests presented in this report have provided an experimental validation of the proposed wind turbine testing methodology, particularly on the wind turbine impedance characterisation and on the evaluation of its steady-state and dynamic performance under different grid conditions.

To the knowledge of the authors a full scale tests with a 4 MW wind turbine have not been reported in the literature, especially where the generator and the converter performance are shown during the same test occasion.

In this paper we have highlighted the test methods for wind turbines by VSC and the theory, methods development with simulations are omitted in this paper.  The theory part are referred to in the background chapter and are well described in the PhD-thesis of the first author Nicolas Espinosa.

Best regards

Ola Carlson and Nicolas Espinosa

[revised manuscript text omitted]